# Upregulation of DJ-1 in Dopaminergic Neurons by a Physically-Modified Saline: Implications for Parkinson’s Disease

**DOI:** 10.3390/ijms24054652

**Published:** 2023-02-28

**Authors:** Malabendu Jana, Sridevi Dasarathy, Supurna Ghosh, Kalipada Pahan

**Affiliations:** 1Department of Neurological Sciences, Rush University Medical Center, Chicago, IL 60612, USA; 2Revalesio Corporation, Tacoma, WA 98421, USA; 3Division of Research and Development, Jesse Brown Veterans Affairs Medical Center, Chicago, IL 60612, USA

**Keywords:** physically modified saline, neurons, DJ-1, CREB

## Abstract

Parkinson’s disease (PD) is the second most common neurodegenerative disorder in human and loss-of-functions DJ-1 mutations are associated with a familial form of early onset PD. Functionally, DJ-1 (PARK7), a neuroprotective protein, is known to support mitochondria and protect cells from oxidative stress. Mechanisms and agents by which the level of DJ-1 could be increased in the CNS are poorly described. RNS60 is a bioactive aqueous solution created by exposing normal saline to Taylor-Couette-Poiseuille flow under high oxygen pressure. Recently we have described neuroprotective, immunomodulatory and promyelinogenic properties of RNS60. Here we delineate that RNS60 is also capable of increasing the level of DJ-1 in mouse MN9D neuronal cells and primary dopaminergic neurons, highlighting another new neuroprotective effect of RNS60. While investigating the mechanism we found the presence of cAMP response element (CRE) in *DJ-1* gene promoter and stimulation of CREB activation in neuronal cells by RNS60. Accordingly, RNS60 treatment increased the recruitment of CREB to the *DJ-1* gene promoter in neuronal cells. Interestingly, RNS60 treatment also induced the enrollment of CREB-binding protein (CBP), but not the other histone acetyl transferase p300, to the promoter of *DJ-1* gene. Moreover, knockdown of CREB by siRNA led to the inhibition of RNS60-mediated DJ-1 upregulation, indicating an important role of CREB in DJ-1 upregulation by RNS60. Together, these results indicate that RNS60 upregulates DJ-1 in neuronal cells via CREB–CBP pathway. It may be of benefit for PD and other neurodegenerative disorders.

## 1. Introduction

Parkinson disease (PD) is an age-related neurodegenerative disorder of the brain [1] that is clinically identified by resting tremor, bradykinesia, rigidity and postural instability [1,2]. Two pathological hallmarks of PD are progressive deterioration of the dopaminergic neurons in the substantia nigra pars compacta (SNpc) and the manifestation of intracytoplasmic inclusions (Lewy bodies) [1]. Recent studies have also described gliosis in the SNpc of PD patients [1]. Although causes for the disease are not well known, studies have identified association of environmental, genetic, and immunological factors with the onset of the disease [3]. At the same time, the discovery of various genes, such as α-synuclein [4], parkin [5], DJ-1 [6], PTEN-induced kinase 1 (PINK-1) [7], HtrA2/Omi gene [8], and leucine-rich repeat kinase 2 (LRRK2) [9] linked to familial forms of PD provides vital hints for the understanding the pathogenesis of the disease.

Oxidative stress is involved in the pathogenesis of many human disorders including PD [10,11,12]. Interestingly, among all the PD-associated genes, DJ-1 is the most important one in providing antioxidant defense [13]. On the one hand, DJ-1 helps a damaged cell to break down superoxide by assisting the transcription of MnSOD [14]. Glutamate cysteine ligase is the rate limiting enzyme of glutathione (GSH) biosynthesis [15]. DJ-1 is known to enrich the level of master antioxidant GSH by upregulating glutamate cysteine ligase [16]. However, mechanisms by which DJ-1 expression could be upregulated is poorly understood and such information is critical for understanding the disease process as well as designing new therapeutic approaches for PD.

RNS60 is produced by allowing normal saline to pass through increased oxygen pressure in Taylor-Couette-Poiseuille (TCP) flow [17,18,19]. Therefore, RNS60 is basically electrokinetically modified saline with elevated oxygen, which contains no active pharmaceutical ingredient. Recently, we have shown that RNS60 inhibits the activation of nuclear factor kappa B (NF-κB) to exhibit anti-inflammatory effects in microglia [18]. RNS60 protects nigral dopaminergic neurons in MPTP mouse model of PD [20] and hippocampal neurons in a transgenic mouse model of AD [21]. RNS60 also increased the expression of *Nrf1*, *Tfam*, *Mcu*, and *Tom20* (genes associated with the biogenesis of mitochondria) to upregulate mitochondrial biogenesis in dopaminergic neuronal cells [22]. Respirometric assays uncovers that RNS60 increases spare glycolytic capacity of oligodendrocytes (OL) under normal culture conditions, while enhancing OL spare respiratory capacity in the presence of a metabolic stress [23]. Here, we delineated that RNS60 was capable of increasing the expression of DJ-1 in mouse MN9D neuronal cells and primary dopaminergic neurons. Furthermore, we demonstrated that RNS60 induced the activation of CREB and that RNS60 increased the level of DJ-1 in neuronal cells via CREB–CBP pathway. Our studies indicate that this physically-altered saline may be of therapeutic benefit for PD and other neurodegenerative disorders in which oxidative stress and mitochondrial anomaly contribute to disease pathogenesis.

## 2. Results

### 2.1. RNS60 Upregulates the Level of DJ-1 in Mouse MN9D Dopaminergic Neuronal Cells

DJ-1 is a neuroprotective protein, which have implications in neurodegenerative disorders like Parkinson’s disease (PD), Alzheimer’s disease (AD) and Huntington’s disease [6,24,25,26,27]. To examine whether RNS60 may be used for the upregulation of DJ-1, MN9D dopaminergic neuronal cells were incubated with different doses of RNS60 for 5 h under serum-free conditions. RT-PCR (Figure 1A) and real-time PCR (Figure 1B) results clearly showed that RNS60 dose-dependently increased the mRNA expression of DJ-1 in MN9D cells. Although at a dose of 2% *v/v*, RNS60 significantly increased the mRNA expression of DJ-1, maximum upregulation was seen at a dose of 10% *v*/*v* RNS60 (Figure 1B). Time course study shows that only 1 h of treatment was enough for RNS60 to stimulate the expression of DJ-1 gene (Figure 1C,D). However, a markedly increased level of *DJ-1* mRNA expression was observed when MN9D cells were treated with RNS60 for 5 h (Figure 1C,D).

Next, we examined whether similar to the increase in *DJ-1* mRNA expression, RNS60 could also increase the level of DJ-1 protein at different doses. As evident from Western blot followed by band quantification, RNS60 upregulated the protein level of DJ-1 in mouse MN9D neuronal cells (Figure 2A,B).

However, no such increase was found with NS (Figure 2A,B). Time-dependent analysis showed that 10% *v/v* RNS60 remained unable to increase the protein level of DJ-1 at 2 h of incubation (Figure 2C,D). However, a significant increase in DJ-1 protein was found at 6 h of incubation with RNS60 with maximum increase seen at 24 h of treatment (Figure 2C,D). Therefore, for further experiments while monitoring DJ-1 protein level, cells were treated with RNS60 for 24 h. To further confirm, we also monitored the upregulation of DJ-1 protein by immunofluorescence followed by calculation of DJ-1 MFI. Since MN9D cells are dopaminergic, we employed double-labeling for tyrosine hydroxylase (TH), marker of dopaminergic neurons, and DJ-1. Consistent to Western blot results, 24 h treatment with RNS60, but not NS, also resulted in the augmentation of DJ-1 protein in MN9D neuronal cells (Figure 2E,F). On the other hand, the level of TH did not change after treatment with either RNS60 or NS (Figure 2E), suggesting the specificity of the effect.

### 2.2. Does RNS60 Upregulate DJ-1 in Mouse Primary Dopaminergic Neurons?

Since RNS60 upregulated DJ-1 in MN9D dopaminergic neuronal cells, next, we examined whether RNS60 could increase DJ-1 in primary dopaminergic neurons. Mouse dopaminergic neurons were incubated with RNS60 and NS for 24 h followed by double-labeling for TH and DJ-1. Similar to MN9D cells, RNS60 at a dose of 5% *v/v* also markedly increased the level of DJ-1 in primary dopaminergic neurons (Figure 3A,B). This effect was specific to RNS60 as NS had no effect on the expression of DJ-1 in neurons (Figure 3A,B). Moreover, both RNS60 and NS had no effect on TH (Figure 3A,C), indicating that RNS60 specifically upregulates DJ-1 in dopaminergic neurons without increasing its signature marker TH.

### 2.3. Activation of cAMP Response Element-Binding Protein (CREB) by RNS60 in MN9D Neuronal Cells

Mechanisms by which the transcription of *DJ-1* gene occurs are poorly understood. Therefore, to understand the mechanism by which RNS60 increased the expression of DJ-1 in neuronal cells, we analyzed the mouse *DJ-1* gene promoter using Mat-Inspector V2.2 search machinery and found the presence of a consensus cAMP response element (CRE) (Figure 4A) that allows the transcription factor CREB to bind.

Therefore, we examined if RNS60 alone was capable of inducing the activation of CREB in MN9D neuronal cells by examining the phosphorylation of CREB. RNS60, but not NS, augmented CREB phosphorylation as portrayed by Western blot of phospho-CREB and total CREB (Figure 4B,C). RNS60 treatment upregulated the level of phospho-CREB, but not total CREB, at different minutes of stimulation.

Although RNS60 upregulated the level of phospho-CREB in MN9D neurons significantly (*p* < 0.05 vs. control) at 5 min of stimulation, maximum upregulation (*p* < 0.01 vs. control) of phospho-CREB was seen at 15 min of RNS60 stimulation (Figure 4B,C). To confirm these results further, we performed double-label immunofluorescence of either phospho-CREB and actin (Figure 4D) or total CREB and actin (Figure 4E) followed by quantification of MFI for either phospho-CREB (Figure 4F) or total CREB (Figure 4G). Consistent to Western blot results, RNS60 increased the level of phospho-CREB (Figure 4D,F), but not total CREB (Figure 4E,G), in MN9D neuronal cells. Upon activation, CREB binds to DNA and therefore, to further reinforce this result, we carried out EMSA in order to check the DNA-binding activity of CREB. As evident from Figure 5, at different minutes (5, 15 and 30) of stimulation, RNS60 treatment markedly led to an increase in DNA-binding activity of CREB. However, RNS60-induced DNA-binding activity of CREB decreased at 60 min of stimulation (Figure 5). Moreover, NS remained unable to stimulate the activation of CREB (Figure 5), signifying the specificity.

### 2.4. RNS60 Stimulates the Employment of CREB to the Promoter of DJ-1 Gene in MN9D Neuronal Cells

Next, to understand if CREB was directly engaged in the transcription of *DJ-1* gene, we investigated the employment of CREB to *DJ-1* gene promoter by ChIP assay. From immunoprecipitates of chromatin fragments of RNS60-treated neuronal cells with antibodies against CREB, we were also able to amplify 306 bp fragments (Figure 6A) from the *DJ-1* promoter corresponding to the CRE (Figure 4A). The p300 and CREB-binding protein (CBP), two important histone acetyl transferases, are known to play an important role in gene transcription [28,29]. Therefore, next, we investigated whether p300 and CBP were also involved in RNS60-driven transcription of the *DJ-1* gene in neuronal cells. As evident from PCR (Figure 6A) and real-time PCR (Figure 6B), RNS60 treatment markedly induced the staffing of CBP to the *DJ-1* gene promoter. In contrast, RNS60 did not induce the recruitment of p300 to the CRE of *DJ-1* gene promoter (Figure 6A,B), suggesting that p300 was not involved in RNS60-mediated transcription of *DJ-1* gene in neuronal cells. Expectedly, similar to the enrolment of CBP and CREB to the CREs, RNS60 treatment was able to employ RNA polymerase to the *DJ-1* gene promoter (Figure 6A,B). These results are specific as we remained unable to detect any amplification product either in control MN9D cells or NS-treated MN9D cells (Figure 6A,B). Furthermore, from the immuno-precipitates obtained with control IgG, we did not observe any amplification product (Figure 6A,B). Together, our results suggest that RNS60-induced transcriptional complex at the *DJ-1* promoter contains CREB, CBP and RNA polymerase (Figure 6C).

### 2.5. RNS60 Needs CREB for the Up-Regulation of DJ-1 in MN9D Neuronal Cells

Next, to explore whether CREB is involved in RNS60-induced increase in DJ-1, we employed the siRNA approach. As expected, CREB siRNA, but not control siRNA, decreased the level of CREB in MN9D neuronal cells (Figure 7A,B). To understand whether the expression of DJ-1 depends on CREB in normal cells, we monitored the level of DJ-1 and found that CREB siRNA, but not control siRNA, decreased level of DJ-1 protein in neuronal cells (Figure 7A,C). This result suggest that DJ-1 expression is dependent on CREB in normal cells. Next, we analyzed RNS60-treated cells. Abrogation of RNS60-mediated up-regulation of DJ-1 by siRNA knockdown of CREB (Figure 7D–F) suggests that RNS60 upregulates the expression of DJ-1 via the CREB signaling pathway.

## 3. Discussion

Reduced glutathione is an important physiological antioxidant, which also functions as a thiol buffer, and DJ-1 is a key molecule in mediating cell survival by positively regulating the biosynthesis of reduced glutathione [16]. In addition, DJ-1 facilitates Nrf2-dependent detoxification pathways [30,31] and induces Akt activity via the suppression of phosphatase and tensin homolog (PTEN) [32,33]. DJ-1 can also function as peroxiredoxin-like peroxidase in vivo in which it can defend mitochondria against oxidative stress [34].

It has been shown that DJ-1 is capable of upregulating the human tyrosine hydroxylase gene by inhibiting sumoylation of pyrimidine tract-binding protein-associated splicing factor [35]. Therefore, upregulation of DJ-1 is considered to be beneficial for the damaged nigrostriatum in PD. Consequently, genetic inactivation of DJ-1 is found to be associated to early onset PD [6] and down-regulation of DJ-1 is seen in brains of sporadic PD patients [36]. Accordingly, delineating molecules to increase the level of DJ-1 in neurons and characterizing associated intracellular pathways that are required for the transduction of signals from the cell surface to the nucleus required for upregulation of *DJ-1* gene are active areas of investigation.

RNS60 is a physically reformed saline that is known to contain charge-stabilized nanobubbles [23,37,38,39]. Due to the turmoil caused by TCP, RNS60 is anticipated to contain charge-stabilized nanostructures consisting of an oxygen nanobubble core surrounded by an electrical double-layer at the liquid/gas edge [18]. However, it does not contain any active pharmaceutical ingredients [23,37,38,39]. Here, we establish that RNS60 is capable of increasing DJ-1 in dopaminergic neurons. RNS60, but not NS, increased the level of DJ-1 in mouse MN9D dopaminergic neuronal cells and primary dopaminergic neurons. Recently we have demonstrated protection of dopaminergic neurons and restoration of locomotor functions in MPTP-challenged mice by RNS60 [20]. Since mitochondrial dysfunction and oxidative stress have been shown to contribute to nigrostriatal degeneration in PD patients and in animal models of PD [40,41], our current results indicate that this DJ-1-upregulating efficacy of RNS60 may participate in RNS60-mediated protection of the nigrostriatum in a mouse model of PD. Moreover, due to implications of DJ-1 in the pathogenesis of PD, increase in DJ-1 in neurons by RNS60 may open up an important avenue whereby RNS60 may decrease nigrostriatal injury in PD.

The intracellular signal transduction cascades required for the transcription of *DJ-1* gene are poorly understood and molecular mechanisms by which RNS60 could upregulate DJ-1 were not known. While analyzing the *DJ-1* gene promoter, we found the presence of a consensus cAMP response element (CRE) in the *DJ-1* promoter. Recently we have demonstrated that RNS60 is capable of inducing the activation of CREB in microglia and oligodendrocytes [18,42], suggesting that RNS60 might employ the CREB pathway for the upregulation of DJ-1. Several lines of evidence presented in this manuscript also demonstrate that RNS60 increases the expression of *DJ-1* gene in neuronal cells via CREB. *First*, RNS60 treatment increased the amount of phospho-CREB, but not total CREB, in neuronal cells. *Second,* RNS60 alone induced the DNA-binding activity of CREB in neuronal cells. *Third,* RNS60 treatment induced the enrollment of CREB to the *DJ-1* gene promoter in neuronal cells. *Fourth,* histone acetyl transferases p300 and CREB-binding protein (CBP) play crucial roles in gene transcription. Interestingly, RNS60 treatment prompted the recruitment of CBP, but not p300, to the *DJ-1* gene promoter (Figure 6C). *Fifth*, siRNA knockdown of CREB abrogated RNS60-mediated upregulation of DJ-1 in neuronal cells.

At present, there is no effective therapy for the treatment of PD. Administration of levodopa/carbidopa and/or an agonist of DA has been the standard treatment for PD symptoms all over the world. However, these medications do not affect the disease course of PD. Moreover, these treatments often lead to side effects and disappointing outcomes. As a result, new effective treatments are necessary. Here, we have proven that RNS60 upregulates the expression nigrostriatum-protecting molecule DJ-1 in dopaminergic neuronal cells via activation of the CREB-CBP signaling pathway. Recently we have demonstrated that RNS60 treatment inhibits glial activation and protects dopaminergic neurons in the SNpc of MPTP mouse model of PD [20]. RNS60 is also capable of exhibiting neuroprotection in animal models of multiple sclerosis [19,43], Alzheimer’s disease [44], and traumatic brain injury [45]. In a phase II multicenter, randomized, double-blind, placebo-controlled trial in amyotrophic lateral sclerosis, RNS60 exhibits positive effects on respiratory and bulbar function [37].

## 4. Materials and Methods

### 4.1. Reagents

Sources of the antibodies are provided in Table 1. Hank’s balanced salt solution (HBSS), fetal bovine serum (FBS), trypsin, and Dulbecco’s modified Eagle’s medium F-12 (DMEM/F-12) were from Mediatech (USA). Antibiotic-antimycotic mixture and poly-D-lysine were obtained from Sigma-Aldrich (St. Louis, MO, USA).

### 4.2. Preparation of RNS60

RNS60 was generated at Revalesio (Tacoma, WA, USA) using Taylor-Couette-Poiseuille (TCP) flow as described before [18,20,43,45]. Briefly, sodium chloride (0.9%) for irrigation, USP pH 5.6 (4.5-7.0, Hospira, Lake Forest, IL), was processed at 4 °C in the presence of oxygen. Chemically, RNS60 is known to contain water, sodium chloride, 50–60 parts/million oxygen, but no active pharmaceutical ingredients. NS, normal saline from the same manufacturing batch was used as control because NS also contacted the same device surfaces as RNS60 and was bottled in the same way. From careful analysis, it was found that RNS60 and NS were chemically identical [18,19]. No difference between RNS60 and NS was also found by liquid chromatography quadrupole time-of-flight mass spectrometric analysis [18]. On the other hand, by using atomic force microscopy (AFM), we observed that RNS60 has a nanobubble composition different from that of NS [18].

### 4.3. MN9D Cells

These cells were obtained from Dr. A. Heller (University of Chicago, Chicago, IL, USA). Cells were maintained in DMEM (Thermo Fisher Scientific, Waltham, MA, USA) supplemented with 10% (*v/v*) heat inactivated FBS, 3.7 g/L NaHCO_3_, 50 U/mL penicillin, and 50 μg/mL streptomycin in an incubator with an atmosphere of 7% CO_2_ at 37 °C. These cells express abundant tyrosine hydroxylase (TH), synthesize dopamine (DA) and also quantitatively release DA [46,47,48].

### 4.4. Isolation of Mouse Primary Dopaminergic Neurons

Mouse primary dopaminergic neurons were isolated as described earlier [22,47,49]. Briefly, nigra was dissected as a thin slice of ventral mesencephalon tissue from E12.5 to 14 days old fetus, which was homogenized with 1 mL of trypsin for 5 min at 37 °C followed by neutralization of trypsin as described [22,47,49]. Then single cell suspension of nigral tissue was prepared that was plated in the poly-d-lysine pre-coated 75 mm flask. Cells were allowed to differentiate fully for 9–10 days before treatment.

### 4.5. Immunoblot Analysis

Western blotting was performed as described earlier [28,50] with some alterations. Briefly, cells were scraped in 1X RIPA buffer and protein was measured using Bradford reagent. Sodium dodecyl sulfate (SDS) sample buffer was added to protein samples and equal amount of protein from each group was electrophoresed on NuPAGE^®^ Novex^®^ 4–12% Bis-Tris gels (Thermo Fisher Scientific, Waltham, MA, USA) followed by transferring proteins onto a nitrocellulose membrane (Bio-Rad, Hercules, CA, USA). The membrane was then washed for 15 min in TBS plus Tween 20 (TBST) and blocked for 1 h in TBST containing BSA. Next, membranes were incubated overnight at 4 °C under shaking conditions with primary antibodies listed in Table 1. The next day, membranes were washed in TBST for 1 h, incubated in secondary antibodies (all 1:10,000; Jackson ImmunoResearch, West Grove, PA, USA) for 1 h at room temperature, washed for one more hour and visualized under the Odyssey^®^ Infrared Imaging System (Li-COR, Lincoln, NE, USA).

### 4.6. Semi-Quantitative RT-PCR Analysis

Total RNA was prepared followed by digestion with DNase for the removal of any contaminating genomic DNA. Semi-quantitative RT-PCR was carried out as described earlier [51,52], using a RT-PCR kit from Clontech. Briefly, 1 µg of total RNA was reverse transcribed using oligo(dT) as primer and MMLV reverse transcriptase (Thermo Fisher Scientific, Waltham, MA, USA). The resulting cDNA was appropriately diluted, and diluted cDNA was amplified. Amplified products were electrophoresed on a 1.8% agarose gels to be pictured by ethidium bromide staining.

DJ-1:Sense: 5′-CCCCGTGCAGTGTAGCCGTG-3′Antisense: 5′-CAGGCCGTCCTTCTCCACGC-3′GAPDH:Sense: 5′-GGTGAAGGTCGGTGTGAACG-3′Antisense: 5′-TTGGCTCCACCCTTCAAGTG-3′.

### 4.7. Real-Time PCR

It was performed using the ABI-Prism7700 sequence detection system (Thermo Fisher Scientific, Waltham, MA, USA) as described earlier [51,52]. The mRNA expressions of respective genes were normalized to the level of GAPDH mRNA. Data were processed by the ABI Sequence Detection System 1.6 software and analyzed by ANOVA.

### 4.8. Immunofluorescence Analysis

It was performed as described earlier [50,53]. Briefly, cover slips containing 100–200 cells/mm^2^ were fixed with 4% paraformaldehyde followed by treatment with cold ethanol and two rinses in phosphate-buffered saline (PBS). Samples were then blocked with 3% bovine serum albumin (BSA) in PBS-Tween-20 (PBST) for 30 min followed by incubation in PBST containing 1% BSA and mouse anti-DJ-1 (1:200) or rabbit anti-TH (1:500). After three washes in PBST (15 min each), slides were further incubated with Cy2 (Jackson ImmunoResearch Laboratories, Inc.). For negative controls, a set of culture slides was incubated under similar conditions without the primary antibodies. The samples were mounted and observed under an Olympus BX41 fluorescence microscope.

### 4.9. Electrophoretic Mobility Shift Assay (EMSA)

MN9D neuronal cells were treated with RNS60 and NS under serum-free condition. At different time periods of treatment, nuclear extracts were prepared to perform EMSA as described previously [54,55] with some modifications. Briefly, IRDye infrared dye end-labeled oligonucleotides containing the consensus CREB DNA-binding sequence were purchased from Licor Biosciences and nuclear extract was incubated with infrared-labeled probe in binding buffer. Samples were separated on a 6% non-denaturing polyacrylamide gel in 0.25× TBE buffer (Tris borate-EDTA) and analyzed by the Odyssey Infrared Imaging System (LI-COR Biosciences, Lincoln, NE, USA).

### 4.10. Chromatin Immunoprecipitation (ChIP) Assay

Recruitment of CREB to *DJ-1* gene promoter was determined by ChIP assay as described earlier [56,57]. Briefly, MN9D neuronal cells were stimulated with RNS60 under serum free conditions for 1 h followed by fixation of cells by adding formaldehyde (1% final concentration). Cross-linked adducts were then resuspended and sonicated. ChIP was performed on the cell lysate by overnight incubation at 4 °C with 2 µg of anti-CREB, anti-CBP, anti-p300, or anti-RNA Polymerase II antibodies followed by incubation with protein G agarose (Santa Cruz Biotechnology, Dallas, TX, USA) for 2 h. The beads were washed and incubated with an elution buffer. To reverse the cross-linking and purify the DNA, precipitates were incubated in a 65 °C incubator overnight and digested with proteinase K. DNA samples were then purified, precipitated, and precipitates were washed with 75% ethanol, air-dried, and resuspended in TE buffer. The following primers were used for amplification of chromatin fragments of mouse *DJ-1* gene.

Sense: 5′-GAGATCTCATTTACCCTGATTTAA-3′Antisense: 5′-GATCCTGATGCTGCTGCACCCACAG-3′

### 4.11. Measurement of Mean Fluorescence Intensity (MFI)

The “measurement module” of the Microsuite V Olympus software (B3 Biological Suite) was used to measure MFI as described by us in several studies [58,59]. Briefly, images were opened in their green channel and after that, measurement module was opened followed by the selection of two parameters including perimeter and MFI. The perimeter was outlined with rectangular box tool and then associated MFI in that given perimeter was automatically calculated.

### 4.12. Statistical Analysis

Statistical analyses were performed with Student’s *t* test (for two-group comparisons) and one-way ANOVA, followed by Tukey’s multiple-comparison tests, as appropriate (for multiple groups comparison), using GraphPad software (GraphPad Prism Version 9.5.1). Data represented as mean ± SD or mean ± SEM as stated in figure legends. A level of *p* < 0.05 was considered statistically significant.

## 5. Conclusions

DJ-1 is a nigral protecting molecule and here, we describe a new neuroprotective property of RNS60, in which this physically-modified saline increases the transcription of *DJ-1* in dopaminergic neuronal cells through CREB-CBP signaling pathway. Although the in vitro state of mouse dopaminergic neurons in culture and its dealing with RNS60 may not truly bear a resemblance to the in vivo condition of these cells in the brain of PD patients, our results delineate a new neuroprotective property of RNS60 and indicate that RNS60 may have therapeutic potential in PD and other mitochondria-related disorders.

## Figures and Tables

**Figure 1 ijms-24-04652-f001:**
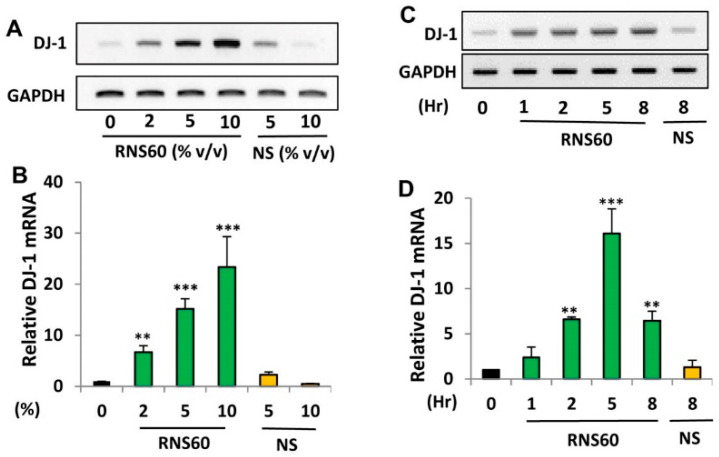
Effect of RNS60 on the expression of *DJ-1* mRNA in mouse MN9D neuronal cells. MN9D cells were treated with different doses (% *v*/*v*) of RNS60 under serum-free condition. NS was used as a negative control. After 5 h of incubation, the level of DJ-1 mRNA was monitored in cells by semi-quantitative RT-PCR (**A**) and real-time PCR (**B**). Cells were treated with 10% *v*/*v* RNS60 under serum-free conditions for different hours followed by analyzing the mRNA expression of DJ-1 by semi-quantitative RT-PCR (**C**) and real-time PCR (**D**). Results are mean ± SD of three different experiments. ** *p* < 0.01 & *** *p* < 0.001 vs. control. GAPDH, glyceraldehyde-3-phosphate dehydrogenase.

**Figure 2 ijms-24-04652-f002:**
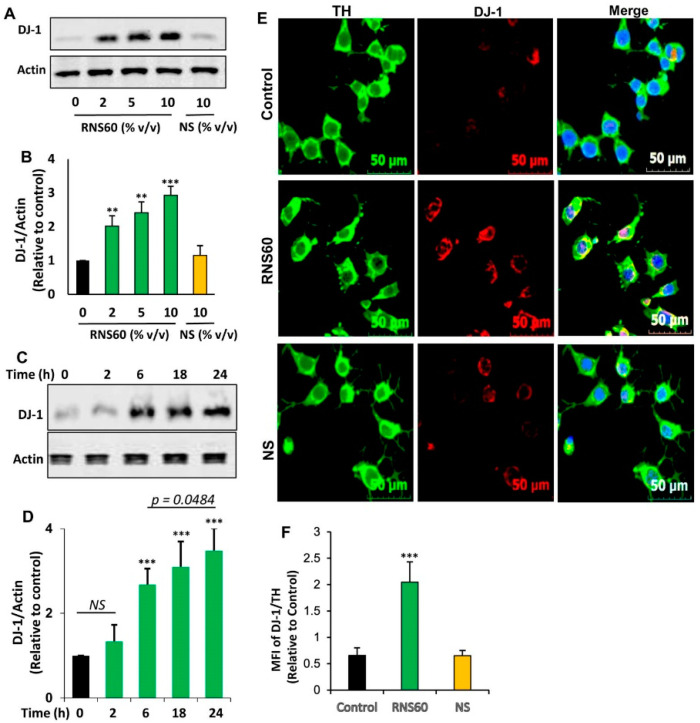
Effect of RNS60 on the expression of DJ-1 protein in mouse MN9D neuronal cells. Cells were treated with different concentrations (% *v/v*) of RNS60 under serum-free condition. NS was used as a negative control. After 24 h of treatment, the level of DJ-1 protein was analyzed by Western blot (**A**). Actin was run as a loading control. Bands were scanned and values (DJ-1/Actin) (**B**) presented as relative to control. Cells were treated with 10 % *v/v* of RNS60 under serum-free condition for different time periods followed by monitoring the level of DJ-1 by Western blot (**C**). Values (DJ-1/Actin) are presented as relative to control (**D**). Results are mean ± SD of three different experiments. ** *p* < 0.01 & *** *p* < 0.001 vs. control. NS, not significant. Cells were double-labeled with antibodies against TH and DJ-1 (**E**). Mean fluorescence intensities (MFI) of DJ-1 and TH were quantified in 10 different images and values (DJ-1/TH) (**F**) presented as relative to control. Data are mean ± SEM of 10 different images per group. *** *p* < 0.001 vs. control.

**Figure 3 ijms-24-04652-f003:**
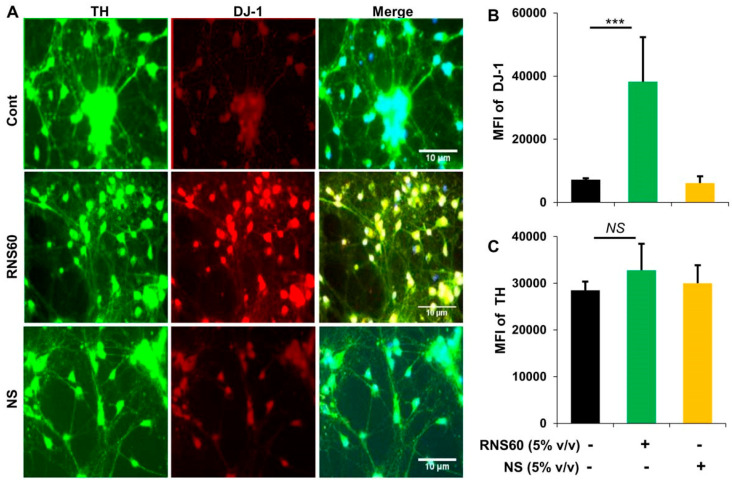
Upregulation of DJ-1 in mouse primary dopaminergic neurons. (**A**) Neurons obtained from ventral mesencephalon of E12.5 to E14 d old fetus received treatment with 5% *v*/*v* RNS60 or NS for 24 h followed by double-labeling with antibodies against DJ-1 and TH. Mean fluorescence intensities (MFI) of DJ-1 (**B**) and TH (**C**) were quantified in 10 different images. Data are mean ± SEM of 10 different images per group. *** *p* < 0.001 vs. control. NS, not significant.

**Figure 4 ijms-24-04652-f004:**
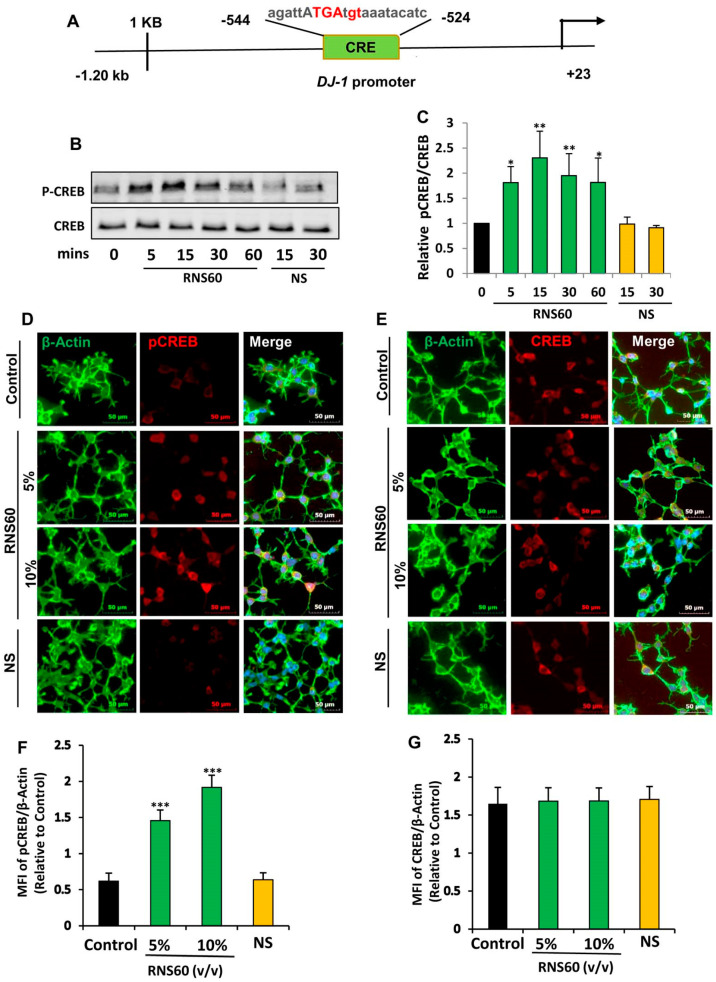
RNS60 induces the activation of CREB in MN9D neuronal cells. (**A**) Analysis of *DJ-1* gene promoter with Mat-Inspector V2.2 search machinery revealed the presence of a consensus cAMP response element (CRE). After different minutes of stimulation with 10% *v/v* RNS60 or NS under serum free conditions, levels of phospho-CREB and total CREB were monitored in cell lysates by Western blotting (**B**). Bands were scanned and values (P-CREB/CREB) are presented as relative to control (**C**). After 15 min of stimulation, cells were double-labeled for either actin & phospho-CREB (**D**) or actin & total CREB (**E**). Mean fluorescence intensities (MFI) were quantified in 10 different images and values ((**F**), actin/phospho-CREB; (**G**), actin/total CREB) presented as relative to control. Results are means ± SD of three different experiments. * *p* < 0.05 & ** *p* < 0.01 & *** *p* < 0.001 vs. control. NS, not significant.

**Figure 5 ijms-24-04652-f005:**
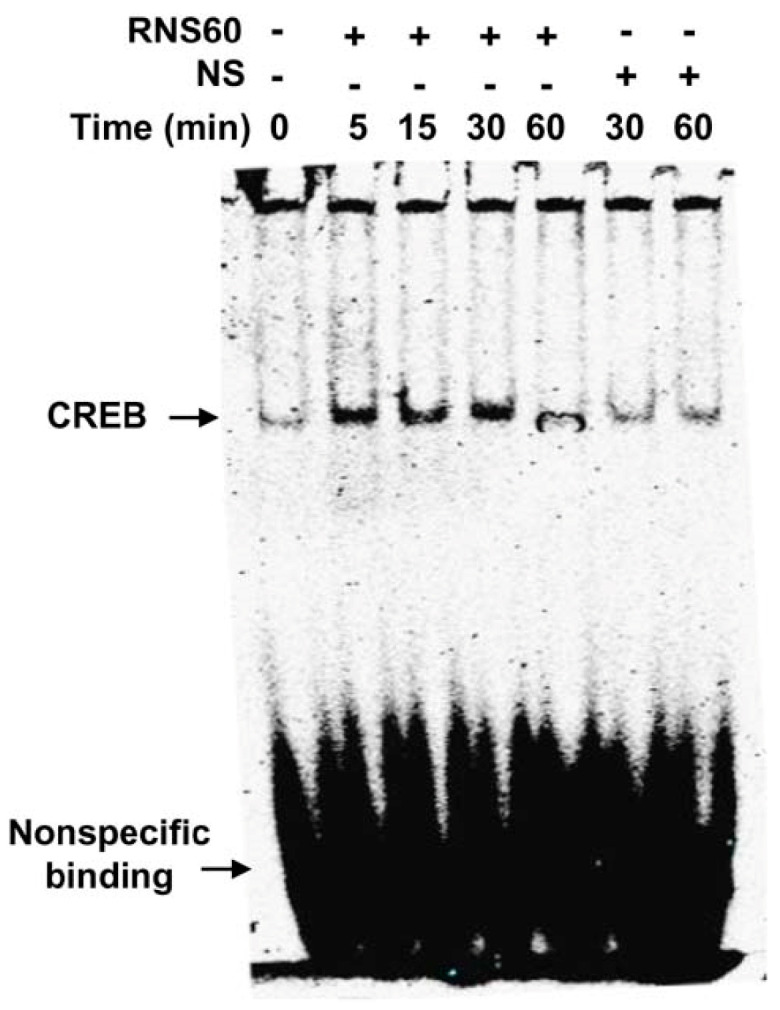
Induction of DNA-binding activity of CREB by RNS60 in MN9D neuronal cells. After different minutes of stimulation with 10% *v/v* RNS60 under serum free conditions, nuclear extracts were examined for EMSA. Cells were stimulated with NS as control. Results represent three independent experiments.

**Figure 6 ijms-24-04652-f006:**
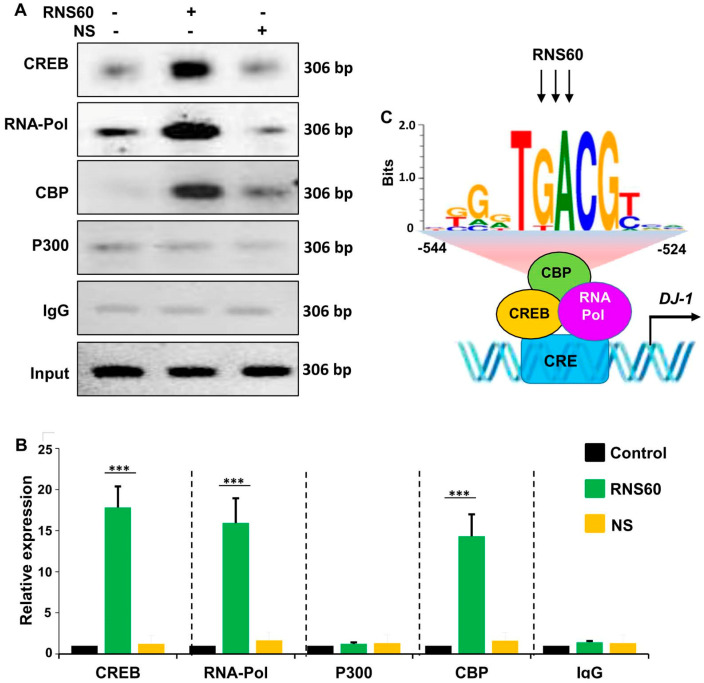
RNS60 treatment stimulates the staffing of CREB to the *DJ-1* gene promoter in MN9D neuronal cells. Cells were incubated with 10% *v/v* RNS60 for 2 h under serum-free condition. Then immunoprecipitated chromatin fragments were amplified by semi-quantitative (**A**) and quantitative (**B**) PCR as described under “Materials and Methods”. Results are the mean ± SD of three separate experiments. *** *p* < 0.001 vs. control. (**C**) A schema depicting RNS60-induced transcriptional activation of the *DJ-1* gene.

**Figure 7 ijms-24-04652-f007:**
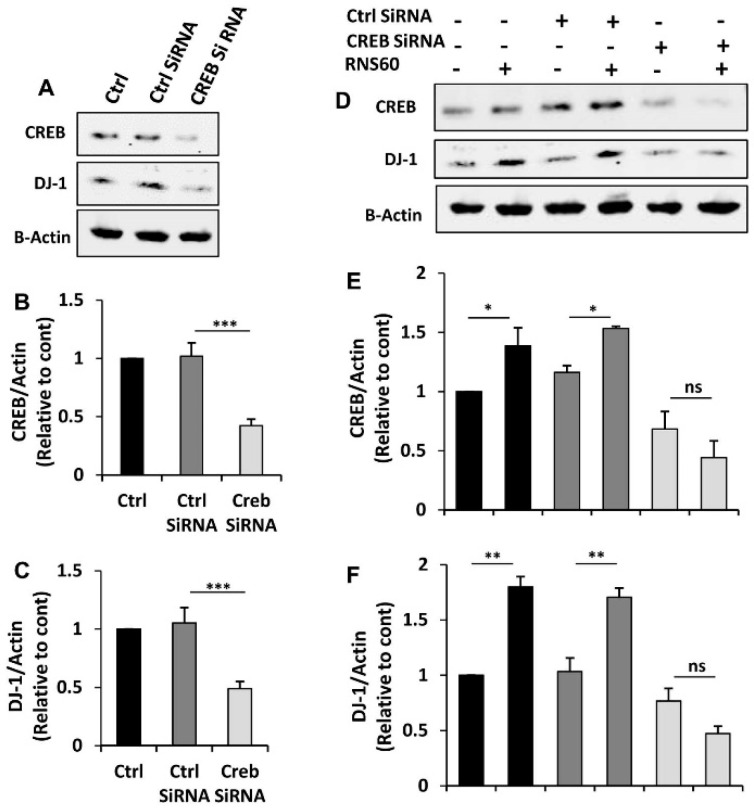
Role of CREB in RNS60-mediated increase in DJ-1 in MN9D neuronal cells. Cells were transfected with CREB siRNA and control siRNA using Lipofectamine Plus and nupherin reagent. To monitor the efficacy of CREB siRNA, after 48 h of transfection, the level of CREB was checked by Western blot (**A**). In parallel blot, the expression of DJ-1 protein was also monitored. Actin was run as a loading control. Bands were scanned and values ((**B**), CREB/Actin; (**C**), DJ-1/Actin) presented as relative to control. Results are mean ± SD of three different experiments. *** *p* < 0.001. After 24 h of transfection, cells were treated with 10% *v/v* RNS60 under serum-free condition for 24 h followed by monitoring the protein levels of CREB and DJ-1 by Western blot (**D**). Actin was run as a loading control. Bands were scanned and values ((**E**), CREB/Actin; (**F**), DJ-1/Actin) presented as relative to control. Results are mean ± SD of three different experiments. * *p* < 0.05; ** *p* < 0.01; ns, not significant.

**Table 1 ijms-24-04652-t001:** Antibodies, sources and dilutions used in this paper.

Antibody	Manufacturer	Catalog	Host	Application Dilution
TH	Abcam	ab137869	Rabbit	WB 1:1000 IF 1:500
DJ-1	Cell Signaling	35743	Mouse	WB 1:1000 IF 1:500
p-CREB	Cell Signaling	9198L	Rabbit	WB 1:500
CREB	Cell Signaling	9197S	Rabbit	WB 1:500
β-actin	Abcam	Ab6276	Mouse	WB 1:6000
CBP	Santa Cruz	SC-369	Rabbit	ChIP 2 µg
p300	Santa Cruz	SC-585	Rabbit	ChIP 2 µg
CREB	Millipore	CS203204	Rabbit	ChIP 2 µg
IgG	Santa Cruz	SC-3888	Rabbit	ChIP 2 µg

IF, immunofluorescence; WB, western blot; ChIP, chromatin immunoprecipitation; TH, tyrosine hydroxylase; CREB, cAMP response element-binding; p-CREB, phospho-CREB; CBP, CREB-binding protein.

## Data Availability

Not applicable.

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
