# Peer review of "Upregulation of DJ-1 in Dopaminergic Neurons by a Physically-Modified Saline: Implications for Parkinson’s Disease"

_ijms, 2023, doi:10.3390/ijms24054652_

Round 1
Reviewer 1 Report
Ref: IJMS manuscript
This manuscript investigates the effects of upregulated DJ-1 in dopaminergic (DA) neurons in Parkinson’s disease. The
work is carried out in a highly experienced investigators’ laboratory who are experts in PD research over many years.
The loss-of- function of DJ-1 (PARK-7), a neuroprotective protein, due to mutation, has been shown to be associated with
familial form of PD at the early onset of the disease. The PARK-7 or DJ-1 is known to support mitochondrial function and protects
cells from oxidative damage. Thus, the focus of the investigation of the manuscript was to understand the various mechanisms
involved in increasing the DJ-1 level in the CNS. Also, to elucidate and examine what other factors may possibly be promoting these increases in DJ-1.
The investigators previous work have shown that bioactive aqueous solution RNS-60, a normal saline generated under greater
oxygen pressure is neuroprotective, immunomodulatory and pro-myelinogenic. Their in vitro studies also demonstrated RNS-60
does upregulate the level of DJ-1 in MN9D neuronal cells and primary DA neurons. From their elaborate investigations they also delineated role of RNS-60 in upregulation of DJ-1 via cAMP response element and its binding protein (CREB). The various
mechanisms in this upregulation of RNS-60 and or DJ-1 are supported by many interesting and important data. The results
are clearly presented. The discussion is pertinent to the results obtained and has clear implications for beneficial effects in PD.
Author Response
Authors highly appreciate enthusiastic comments by Reviewer 1.
Reviewer 2 Report
This manuscript “Upregulation of DJ-1 in dopaminergic neurons by a physically- 2 modified saline: Implications for Parkinson’s disease” by Dr. Pahan’s lab deals with the identification of a new reagent RNS60, which contains charge-stabilized nonobubles, capable of upregulating DJ-1 in dopaminergic neurons. Authors argue, based on the evidence of involvement of DJ-1 in the pathogenesis of PD, upregulation of DJ-1 in neurons by RNS60 may be an important reagent to decrease nigrostriatal injury in PD. Parkinson’s disease is progressive disabling disease. It is estimated that 60-80 thousand new cases are diagnosed each year, joining millions of Americans who currently have PD, and there is no cure for this disease. PD is characterized by the presence of degenerating dopaminergic neurons, Lewy bodies and activated microglia in brain. At present, no effective therapy is available for PD. Administration of a DA agonist or levodopa has been the standard treatment for PD symptoms but does not affect the disease course. Moreover, these treatments often lead to side effects and unsatisfactory outcomes. As a result, new effective treatments are necessary.
Results presented in this manuscript clearly show increases in the expression of DJ-1 gene in neuronal cells via preferential increase in phospho-CREB by RNS60 treatment. Results also show that RNS60 treatment induced the DNA-binding activity of CREB in neuronal cells along with the enrollment of CREB to the DJ-1 gene promoter in neuronal cells. Additionally, RNS60 treatment prompted the recruitment of CREB-binding protein (CBP), which plays crucial roles in gene transcription, to the DJ-1 gene promoter.
The paper is well-written. The methodology is sound, the results are convincing and clearly presented, and the discussion is thorough and complete. The data are highly relevant to the understanding of brain function. The results have strong implications on neuroprotection and its beneficial effects on neurodegenerative diseases such as PD.
Author Response
Thank you very much for nice comments.
Reviewer 3 Report
Parkinson disease is the second most prevalent neurodegenerative disorders after Alzheimer’s disease and currently there is no cure for it. In this study, the authors showed that RNS60 has the ability to increase the mRNA and protein levels of DJ-1 in mouse MN9D neuronal cells and primary dopaminergic neurons. They discovered the cAMP response element (CRE) in the DJ-1 gene promoter and went further to demonstrate that the effect on DJ-1 expression is through the recruitment of the CREB/CBP, but not p300, to the DJ-1 gene promoter. The manuscript is interesting and has the potential to reach higher quality.
Here are my concerns and comments.
1. Cell images in all figures are extremely low in resolution and low in quality. Even the scale bars are barely visible. I am also puzzled by the low resolution of other non-image figures. (This could be caused by the loss of resolution while generating PDF file for review.)
2. Time course of RNS60 effect on the transcription and on translation of DJ-1. The authors reported the DJ-1 mRNA expression after RNS60 treatment and observed maximum effect at 5 hours. No similar time course observation was done to observe the RNS60 effect on the DJ-1 protein levels. Rather we see a snapshot at 24 hour point.
3. Figure 1: should appear after “Results” section title. Furthermore, it should appear in two columns: A and B in one column; and C and D in the other.
4. Figure 6: Labeling in Panel A does not make sense.
5. Line 44: “In one hand” should be “On the one hand”.
6. Line 416, “Data Availability Statement”: “Please refer to suggested Data Availability Statements in section 416 “MDPI Research Data Policies” at https://www.mdpi.com/ethics. 417” Please rework or remove.
Author Response
Please see our response below:
Comment 1: Cell images in all figures are extremely low in resolution and low in quality. Even the scale bars are barely visible. I am also puzzled by the low resolution of other non-image figures. (This could be caused by the loss of resolution while generating PDF file for review.)
Response: We have taken care of this.
Comment 2: Time course of RNS60 effect on the transcription and on translation of DJ-1. The authors reported the DJ-1 mRNA expression after RNS60 treatment and observed maximum effect at 5 hours. No similar time course observation was done to observe the RNS60 effect on the DJ-1 protein levels. Rather we see a snapshot at 24 hour point.
Response: We have performed this experiment. Please see Figure 2C-D.
Comment 3: Figure 1: should appear after “Results” section title. Furthermore, it should appear in two columns: A and B in one column; and C and D in the other.
Response: We have done this. Thank you.
Comment 4: Figure 6: Labeling in Panel A does not make sense.
Response: We have fixed this. Thank you.
Comment 5: Line 44: “In one hand” should be “On the one hand”.
Response: We have taken care of this. Thank you.
Comment 6: Line 416, “Data Availability Statement”: “Please refer to suggested Data Availability Statements in section 416 “MDPI Research Data Policies” at https://www.mdpi.com/ethics. 417” Please rework or remove.
Response: We have taken care of this. Thank you.
Round 2
Reviewer 3 Report
I am very pleased to see that the quality of the images are significant improved. Now I fully support the publication of the revised manuscript.